# ADAPTIVE NORMS FOR DEEP LEARNING WITH REGULARIZED NEWTON METHODS

## ABSTRACT

We investigate the use of regularized Newton methods with adaptive norms for optimizing neural networks. This approach can be seen as a second-order counterpart of adaptive gradient methods, which we here show to be interpretable as first-order trust region methods with ellipsoidal constraints. In particular, we prove that the preconditioning matrix used in RMSProp and Adam satisfies the necessary conditions for provable convergence of second-order trust region methods with standard worst-case complexities on general non-convex objectives. Furthermore, we run experiments across different neural architectures and datasets to find that the ellipsoidal constraints constantly outperform their spherical counterpart both in terms of number of backpropagations and asymptotic loss value. Finally, we find comparable performance to state-of-the-art first-order methods in terms of backpropagations, but further advances in hardware are needed to render Newton methods competitive in terms of computational time.

## 1 INTRODUCTION

We consider finite-sum optimization problems of the form

$$\min_{\mathbf{w} \in \mathbb{R}^d} \left[ \mathcal{L}(\mathbf{w}) := \sum_{i=1}^{n} \ell(f(\mathbf{w}, \mathbf{x}_i, \mathbf{y}_i)) \right], \qquad (1)$$

which typically arise in neural network training, e.g. for empirical risk minimization over a set of data points $(\mathbf{x}_i, \mathbf{y}_i) \in \mathbb{R}^{in} \times \mathbb{R}^{out}, i = 1, \ldots, n$. Here, $\ell : \mathbb{R}^{out} \times \mathbb{R}^{out} \to \mathbb{R}^+$ is a convex loss function and $f : \mathbb{R}^{in} \times \mathbb{R}^d \to \mathbb{R}^{out}$ represents the neural network mapping parameterized by the concatenation of the weight layers $\mathbf{w} \in \mathbb{R}^d$, which is non-convex due to its multiplicative nature and potentially non-linear activation functions. We assume that $\mathcal{L}$ is lower bounded and twice differentiable, i.e. $\mathcal{L} \in C^2(\mathbb{R}^d, \mathbb{R})$ and consider finding a first- and second-order stationary point $\bar{\mathbf{w}}$ for which $\|\nabla \mathcal{L}(\bar{\mathbf{w}})\| \leq \epsilon_g$ and $\lambda_{\min}(\nabla^2 \mathcal{L}(\bar{\mathbf{w}})) \geq -\epsilon_H$.

In the era of deep neural networks, stochastic gradient descent (SGD) is one of the most widely used training algorithms (Bottou, 2010). What makes SGD so attractive is its simplicity and per-iteration cost that is independent of the size of the training set ($n$) and scale linearly in the dimensionality ($d$). However, gradient descent is known to be inadequate to optimize functions that are ill-conditioned (Nesterov, 2013; Shalev-Shwartz et al., 2017) and thus adaptive gradient methods that employ dynamic, coordinate-wise learning rates based on past gradients—including Adagrad (Duchi et al., 2011), RMSprop (Tieleman & Hinton, 2012) and Adam (Kingma & Ba, 2014)—have become a popular alternative, often providing significant speed-ups over SGD.

From a theoretical perspective, Newton methods provide stronger convergence guarantees by appropriately transforming the gradient in ill-conditioned regions according to second-order derivatives. It is precisely this Hessian information that allows *regularized* Newton methods to enjoy superlinear local convergence as well as to provably escape saddle points (Conn et al., 2000). While second-order algorithms have a long-standing history even in the realm of neural network training (Hagan & Menhaj, 1994; Becker et al., 1988), they were mostly considered as too computationally and memory expensive for practical applications. Yet, the seminal work of Martens (2010) renewed interest for their use in deep learning by proposing efficient *Hessian-free* methods that only access second-order information via matrix-vector products which can be computed at the cost of an additional backpropagation (Pearlmutter, 1994; Schraudolph, 2002). Among the class of regularized Newton methods,

trust region (Conn et al., 2000) and cubic regularization algorithms (Cartis et al., 2011) are the most principled approaches in the sense that they yield the strongest convergence guarantees. Recently, stochastic extensions have emerged (Xu et al., 2017b; Yao et al., 2018; Kohler & Lucchi, 2017; Gratton et al., 2017), which suggest their applicability for deep learning.

We here propose a simple modification to make TR methods even more suitable for neural network training. Particularly, we build upon the following alternative view on adaptive gradient methods:

*While gradient descent can be interpreted as a spherically constrained first-order TR method, preconditioned gradient methods—such as Adagrad—can be seen as first-order TR methods with ellipsoidal trust region constraint.*

This observation is particularly interesting since spherical constraints are blind to the underlying geometry of the problem, but ellipsoids can adapt to local landscape characteristics, thereby allowing for more suitable steps in regions that are ill-conditioned. We will leverage this analogy and investigate the use of the Adagrad and RMSProp preconditioning matrices as *ellipsoidal* trust region shapes within a stochastic second-order TR algorithm (Xu et al., 2017a; Yao et al., 2018). Since no ellipsoid fits all objective functions, our main contribution lies in the identification of adequate matrix-induced constraints that lead to provable convergence and significant practical speed-ups for the specific case of deep learning. On the whole, our contribution is threefold:

- We provide a new perspective on adaptive gradient methods that contributes to a better understanding of their inner-workings.
- We investigate the first application of ellipsoidal TR methods for deep learning. We show that the RMSProp matrix can directly be applied as constraint inducing norm in second-order TR algorithms while *preserving all convergence guarantees* (Theorem 1).
- Finally, we provide an experimental benchmark across different real-world datasets and architectures (Section 5). We compare second-order methods also to adaptive gradient methods and show results in terms of backpropagations, epochs, and wall-clock time; a comparison we were not able to find in the literature.

Our main empirical results demonstrate that ellipsoidal constraints prove to be a very effective modification of the trust region method in the sense that they constantly outperform the spherical TR method, both in terms of number of backprogations and asymptotic loss value on a variety of tasks.

## 2 RELATED WORK

**First-order methods** The prototypical method for optimizing Eq. (1) is SGD (Robbins & Monro, 1951). The practical success of SGD in non-convex optimization is unquestioned and theoretical explanations of this phenomenon are starting to appear. Recent findings suggest the ability of this method to escape saddle points and reach local minima in polynomial time, but they either need to artificially add noise to the iterates (Ge et al., 2015; Lee et al., 2016) or make an assumption on the inherent noise of SGD (Daneshmand et al., 2018). For neural networks, a recent line of research proclaims the effectiveness of SGD, but the results come at the cost of strong assumptions such as heavy over-parametrization and Gaussian inputs (Du et al., 2017; Brutzkus & Globerson, 2017; Li & Yuan, 2017; Du & Lee, 2018; Allen-Zhu et al., 2018). Adaptive gradient methods (Duchi et al., 2011; Tieleman & Hinton, 2012; Kingma & Ba, 2014) build on the intuition that larger (smaller) learning rates for smaller (larger) gradient components balance their respective influences and thereby the methods behave as if optimizing a more isotropic surface. Such approaches have first been suggested for neural nets by LeCun et al. (2012) and convergence guarantees are starting to appear (Ward et al., 2018; Li & Orabona, 2018). However, these are not superior to the $\mathcal{O}(\epsilon_g^{-2})$ worst-case complexity of standard gradient descent (Cartis et al., 2012b).

**Regularized Newton methods** The most principled class of regularized Newton methods are trust region (TR) and adaptive cubic regularization algorithms (ARC) (Conn et al., 2000; Cartis et al., 2011), which repeatedly optimize a local Taylor model of the objective while making sure that the step does not travel too far such that the model stays accurate. While the former finds first-order stationary points within $\mathcal{O}(\epsilon_g^{-2})$, ARC only takes at most $\mathcal{O}(\epsilon_g^{-3/2})$. However, simple modifications to the TR framework allow these methods to obtain the same accelerated rate (Curtis et al., 2017). Both methods

take at most $\mathcal{O}(\epsilon_H^{-3})$ iterations to find an $\epsilon_H$ approximate second-order stationary point (Cartis et al., 2012a). These rates are optimal for second-order Lipschitz continuous functions (Carmon et al., 2017; Cartis et al., 2012a) and they can be retained even when only sub-sampled gradient and Hessian information is used (Kohler & Lucchi, 2017; Yao et al., 2018; Xu et al., 2017b; Blanchet et al., 2016; Liu et al., 2018). Furthermore, the involved Hessian information can be computed solely based on Hessian-vector products, which are implementable very efficiently (Pearlmutter, 1994). This makes these methods particularly attractive for deep learning, but the empirical evidence of their applicability is rather limited. We are only aware of the works of Liu et al. (2018) and Xu et al. (2017a), which report promising first results but are by no means fully encompassing.

**Gauss-Newton methods** An interesting line of research proposes to replace the Hessian by (approximations of) the generalized-Gauss-Newton matrix (GGN) within a Levenberg-Marquardt framework[1] (LeCun et al., 2012; Martens, 2010; Martens & Grosse, 2015). As the GGN matrix is always positive semidefinite, these methods cannot leverage negative curvature to escape saddles and hence, there exist no second-order convergence guarantees. Furthermore, there are cases in neural networks where the Hessian is better conditioned than the GGN matrix (Mizutani & Dreyfus, 2008). Nevertheless, the above works report promising preliminary results, most notably Grosse & Martens (2016) find that K-FAC can be faster than SGD on a small convnet. On the other hand, recent findings report performance at best comparable to SGD on the much larger ResNet architecture (Ma et al., 2019). Moreover, Xu et al. (2017a) reports many cases where TR and GGN algorithms perform similarly. This line of work can be seen as complementary to our approach since it is straightforward to replace the Hessian in the TR framework with the GGN matrix. Furthermore, the preconditioners used in Martens (2010) and Chapelle & Erhan (2011), namely diagonal estimates of the empirical Fisher and Fisher matrix, respectively, can directly be used as matrix norms in our ellipsoidal TR framework.

## 3 An Alternative View on Adaptive Gradient Methods

Adaptively preconditioned gradient methods update iterates as $\mathbf{w}_{t+1} = \mathbf{w}_t - \eta_t \mathbf{A}_t^{-1/2} \mathbf{g}_t$, where $\mathbf{g}_t$ is a stochastic estimate of $\nabla \mathcal{L}(\mathbf{w}_t)$ and $\mathbf{A}_t$ is a positive definite symmetric pre-conditioning matrix. In Adagrad, $\mathbf{A}_{ada,t}$ is the un-centered second moment matrix of the past gradients computed as

$$\mathbf{A}_{ada,t} := \mathbf{G}_t \mathbf{G}_t^\mathsf{T} + \epsilon \mathbf{I}, \tag{2}$$

where $\epsilon > 0$, $\mathbf{I}$ is the $d \times d$ identity matrix and $\mathbf{G}_t = [\mathbf{g}_1, \mathbf{g}_2, \dots, \mathbf{g}_t]$. Building up on the intuition that past gradients might become obsolete in quickly changing non-convex landscapes, RMSprop (and Adam) introduce an exponential weight decay leading to the preconditioning matrix

$$\mathbf{A}_{rms,t} := \big((1-\beta)\mathbf{G}_t \operatorname{diag}(\beta^t, \dots, \beta^0)\mathbf{G}_t^\mathsf{T}\big) + \epsilon \mathbf{I}, \tag{3}$$

where $\beta \in (0, 1)$. In order to save computational efforts, the diagonal versions $\operatorname{diag}(\mathbf{A}_{ada})$ and $\operatorname{diag}(\mathbf{A}_{rms})$ are more commonly applied in practice, which in turn gives rise to coordinate-wise adaptive stepsizes that are enlarged (reduced) in coordinates that have seen past gradient components with a smaller (larger) magnitude.

### 3.1 Adaptive preconditioning as ellipsoidal Trust Region

Starting from the fact that adaptive methods employ coordinate-wise stepsizes, one can take a principled view on these methods. Namely, their update steps arise from minimizing a first-order Taylor model of the function $\mathcal{L}$ within an *ellipsoidal* search space around the current iterate $\mathbf{w}_t$, where the diameter of the ellipsoid along a particular coordinate is *implicitly* given by $\eta_t$ and $\|\mathbf{g}_t\|_{\mathbf{A}_t^{-1}}$. Correspondingly, vanilla (S)GD optimizes the same first-order model within a *spherical* constraint. Figure 1 (top) illustrates this effect by showing not only the iterates of GD and Adagrad but also the implicit trust regions within which the local models were optimized at each step.[2]

It is well known that GD struggles to progress towards the minimizer of quadratics along low-curvature directions (see e.g., Goh (2017)). While this effect is negligible for well-conditioned objectives (Fig. 1, left), it leads to a drastic slow-down when the problem is ill-conditioned (Fig. 1,

---

[1]This algorithm is a simplified TR method, initially tailored for non-linear least squares problems (Nocedal & Wright, 2006)

[2]We only plot every other trust region. Since the models are linear, the minimizer is always on the boundary.

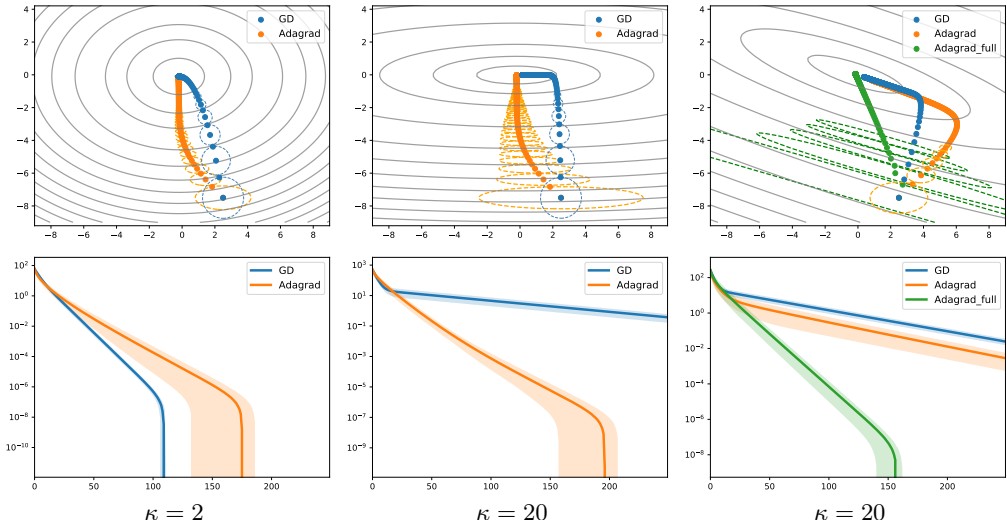

Figure 1: Top: Iterates and implicit trust regions of GD and Adagrad on quadratic objectives with different condition number $\kappa$. Bottom: Average log suboptimality over iterations as well as 90% confidence intervals of 30 runs with random initialization

center). Particularly, once the method has reached the bottom of the valley, it struggles to make progress along the horizontal axis. Here is precisely where the advantage of adaptive stepsize methods comes into play. As illustrated by the dashed lines, Adagrad's search space is damped along the direction of high curvature (vertical axis) and elongated along the low curvature direction (horizontal axis). This allows the method to move further horizontally early on to enter the valley with a smaller distance to the optimizer $\mathbf{w}^*$ along the low curvature direction which accelerates convergence.

Let us now formally establish the result that allows us to re-interpret adaptive gradient methods from the trust region perspective introduced above.

---

**Lemma 1** (Preconditioned gradient methods as TR). *A preconditioned gradient step*

$$\mathbf{w}_{t+1} - \mathbf{w}_t = \mathbf{s}_t := -\eta_t \mathbf{A}_t^{-1} \mathbf{g}_t \tag{4}$$

*with stepsize $\eta_t > 0$, symmetric positive definite preconditioner $\mathbf{A}_t \in \mathbb{R}^{d \times d}$ and $\mathbf{g}_t \neq 0$ minimizes a first-order model around $\mathbf{w}_t \in \mathbb{R}^d$ in an ellipsoid given by $\mathbf{A}_t$ in the sense that*

$$\mathbf{s}_t := \arg\min_{\mathbf{s} \in \mathbb{R}^d} \left[ m_t^1(\mathbf{s}) = \mathcal{L}(\mathbf{w}_t) + \mathbf{s}^\intercal \mathbf{g}_t \right], \quad s.t. \quad \|\mathbf{s}\|_{\mathbf{A}_t} \leq \eta_t \|\mathbf{g}_t\|_{\mathbf{A}_t^{-1}}. \tag{5}$$

---

**Corollary 1** (Rmsprop). *The step $\mathbf{s}_{rms,t} := -\eta_t \mathbf{A}_{rms,t}^{-1/2} \mathbf{g}_t$ minimizes a first-order Taylor model around $\mathbf{w}_t$ in an ellipsoid given by $\mathbf{A}_{rms,t}^{1/2}$ (Eq. 3) in the sense that*

$$\mathbf{s}_{rms,t} := \arg\min_{\mathbf{s} \in \mathbb{R}^d} \left[ m_t^1(\mathbf{s}) = \mathcal{L}(\mathbf{w}_t) + \mathbf{s}^\intercal \mathbf{g}_t \right], \quad s.t. \quad \|\mathbf{s}\|_{\mathbf{A}_{rms,t}^{1/2}} \leq \eta_t \|\mathbf{g}_t\|_{\mathbf{A}_{rms,t}^{-1/2}}. \tag{6}$$

Equivalent results can be established for Adam using $\mathbf{g}_{adam,t} := (1-\beta) \sum_{k=0}^t \beta^{t-k} \mathbf{g}_t$ as well as for Adagrad by replacing the matrix $\mathbf{A}_{ada}$ into the constraint in Eq. (6). Of course, the update procedure in Eq. (5) is merely a reinterpretation of the original preconditioned update, and thus the employed trust region radii are defined *implicitly* by the current gradient and stepsize.

## 3.2 DIAGONAL VERSUS FULL PRECONDITIONING

A closer look at Figure 1 reveals that the first two problems are perfectly *axis-aligned*, which makes these objectives particularly attractive for diagonal preconditioning. For comparison, we report another quadratic instance, where the Hessian is no longer zero on the off-diagonals (Fig. 1, right). As can be seen, this introduces a tilt in the level sets and reduces the superiority of diagonal Adagrad

over plain GD. However, using the full preconditioner $\mathbf{A}_{ada}$ re-establishes the original speed up. Yet, non-diagonal preconditioning comes at the cost of taking the inverse square root of a large matrix, which is why this approach has been relatively unexplored (see Agarwal et al. (2018) for an exception). Interestingly, early results by Becker et al. (1988) on the curvature of neural nets report a strong diagonal dominance of the Hessian matrix $\nabla^2 \mathcal{L}(\mathbf{w})$. However, the reported numbers are only for tiny networks of at most 256 parameters. We here take a first step towards generalizing these findings to modern day networks. Furthermore, we contrast the diagonal dominance of real Hessians to the expected behavior of random Wigner matrices.[3] For further evidence, we also compare Hessians of Ordinary Least Squares (OLS) problems with random inputs. For this purpose, let $\delta_{\mathbf{A}}$ define the ratio of diagonal to overall mass of a matrix $\mathbf{A}$, i.e. $\delta_{\mathbf{A}} := \frac{\sum_i |\mathbf{A}_{i,i}|}{\sum_i \sum_j |\mathbf{A}_{i,j}|}$ as in (Becker et al., 1988).

**Proposition 1** (Diagonal share of Wigner matrix). *For a random Gaussian[4] Wigner matrix $\mathbf{W}$ (see Eq. (42)) the diagonal mass of the expected absolute matrix amounts to:* $\delta_{\mathbb{E}[|\mathbf{W}|]} = \frac{1}{1+(d-1)\frac{\sigma_2}{\sigma_1}}$.

Thus, if we suppose the Hessian at any given point $\mathbf{w}$ were a random Wigner matrix we would expect the share of diagonal mass to fall with $\mathcal{O}(1/d)$ as the network grows in size. In the following, we derive a similar result for the large $n$ limit in the case of OLS Hessians.

**Proposition 2** (Diagonal share of OLS Hessian). *Let $\mathbf{X} \in \mathbb{R}^{d \times n}$ and assume each $\mathbf{x}_{i,j}$ is generated i.i.d. with zero-mean and finite second moment $\sigma^2 > 0$. Then the share of diagonal mass of the expected matrix $\mathbb{E}[|\mathbf{H}_{ols}|]$ amounts to:* $\delta_{\mathbb{E}[|\mathbf{H}_{ols}|]} \overset{n \to \infty}{\to} \frac{\sqrt{n}}{\sqrt{n}+(d-1)\sqrt{\frac{2}{\pi}}}$.

Empirical simulations suggest that this result holds already in small $n$ settings (see Figure D.2) and finite $n$ results can be likely derived under assumptions such as Gaussian data. As can be seen in Figure 2 below, even for a practical batch size of $n = 32$ the diagonal mass $\delta_{\mathbf{H}}$ of neural networks stays above both benchmarks for random inputs as well as with real-world data.

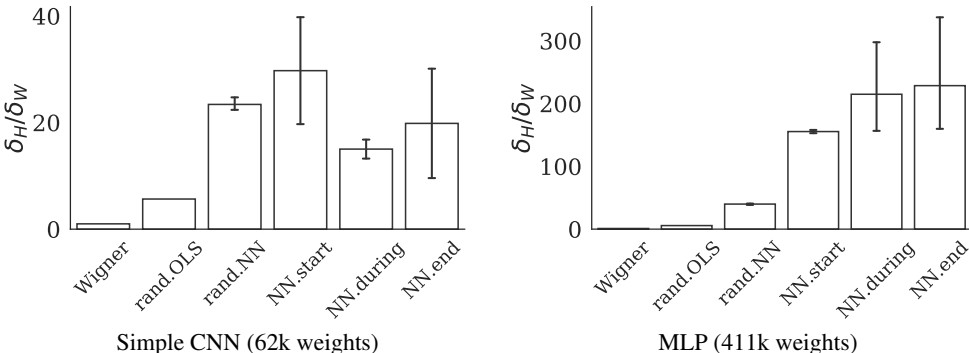

Figure 2: Diagonal mass of neural network Hessian $\delta_{\mathbf{H}}$ relative to $\delta_{\mathbb{E}[|\mathbf{W}|]}$ and $\delta_{\mathbb{E}[|\mathbf{H}_{ols}|]}$ of corresponding dimensionality for random inputs as well as at random initialization, middle and after reaching 90% training accuracy with RMSProp on CIFAR-10. Mean and 95% confidence interval over 10 independent runs.

These results are in line with Becker et al. (1988) and suggest that full matrix preconditioning might indeed not be worth the additional computational cost. We thus use diagonal preconditioning in all of our experiments in Section 5 but note that further theoretical and empirical elaborations of these findings are needed to assess the Hessian structure and hence effectiveness of full-matrix pre-conditioning, which is out of the scope of the work at hand.

## 4 SECOND-ORDER TRUST REGION METHODS

Cubic regularization (Nesterov & Polyak, 2006; Cartis et al., 2011) and trust region methods belong to the family of globalized Newton methods. Both frameworks compute parameter updates by

---

[3]Of course, Hessians do not have i.i.d. entries but the symmetry of Wigner matrices suggests that this baseline is not completely off.

[4]The argument naturally extends to any distribution with positive expected absolute values.

optimizing regularized (former) or constrained (latter) second-order Taylor models of the objective $\mathcal{L}$ around the current iterate $\mathbf{w}_t$.[5] In particular, in iteration $t$ the update step of the trust region algorithm is computed as

$$\min_{\mathbf{s} \in \mathbb{R}^d} \left[ m_t(\mathbf{s}) := \mathcal{L}(\mathbf{w}_t) + \mathbf{g}_t^\mathsf{T} \mathbf{s} + \frac{1}{2} \mathbf{s}^\mathsf{T} \mathbf{B}_t \mathbf{s} \right], \text{ s.t. } \|\mathbf{s}\|_{\mathbf{A}_t} \leq \Delta_t, \tag{7}$$

where $\Delta_t > 0$ and $\mathbf{g}_t$ and $\mathbf{B}_t$ are either $\nabla \mathcal{L}(\mathbf{w}_t)$ and $\nabla^2 \mathcal{L}(\mathbf{w}_t)$ or suitable approximations. The matrix $\mathbf{A}_t$ induces the shape of the constraint set. So far, the common choice for neural networks is $\mathbf{A}_t := \mathbf{I}$, $\forall t$ which gives rise to spherical trust regions (Xu et al., 2017a; Liu et al., 2018). By solving the *constrained* problem (7), TR methods overcome the problem that pure Newton steps may be ascending, attracted by saddles or not even computable. Please see Appendix B for more details.

**Why ellipsoids?** There are many sources for ill-conditioning in neural networks such as un-centered and correlated inputs (LeCun et al., 2012), saturated hidden units, and different weight scales in different layers (Van Der Smagt & Hirzinger, 1998). While the quadratic term of model (7) accounts for such ill-conditioning to some extent, the spherical constraint is completely blind towards the loss surface. Thus, it is advisable to instead measure distances in norms that reflect the underlying geometry (see Chap. 7.7 in Conn et al. (2000)). The ellipsoids we propose are such that they allow for longer steps along coordinates that have seen small gradient components in the past and vice versa. Thereby the TR shape is adaptively adjusted to fit the current region of the loss landscape. This is not only effective when the iterates are in an ill-conditioned neighborhood of a minimizer (Fig. 1), but it also helps to escape elongated plateaus (see autoencoder in Sec. 5). Contrary to adaptive first-order methods, the diameter ($\Delta_t$) is updated directly depending on whether or not the local Taylor model is an adequate approximation at the current point.

### 4.1 CONVERGENCE OF ELLIPSOIDAL TRUST REGION METHODS

Inspired by the success of adaptive gradient methods, we investigate the use of their preconditioning matrices as norm inducing matrices for second-order TR methods. The crucial condition for convergence is that the applied norms are not degenerate during the entire minimization process in the sense that the ellipsoids do not flatten out (or blow up) completely along any given direction. The following definition formalizes this intuition.

**Definition 1** (Uniformly equivalent norms). The norms $\|\mathbf{w}\|_{\mathbf{A}_t} := (\mathbf{w}^\mathsf{T} \mathbf{A}_t \mathbf{w})^{1/2}$ induced by symmetric positive definite matrices $\mathbf{A}_t$ are called uniformly equivalent, if $\exists \mu \geq 1$ such that $\forall \mathbf{w} \in \mathbb{R}^d, \forall t = 1, 2, \dots$

$$\frac{1}{\mu} \|\mathbf{w}\|_{\mathbf{A}_t} \leq \|\mathbf{w}\|_2 \leq \mu \|\mathbf{w}\|_{\mathbf{A}_t}. \tag{8}$$

We now establish a result which shows that the RMSProp ellipsoid is indeed uniformly equivalent.

---

**Lemma 2** (Uniform equivalence). *Suppose $\|\mathbf{g}_t\|^2 \leq L_H^2$ for all $\mathbf{w}_t \in \mathbb{R}^d$, $t = 1, 2, \dots$ Then there always exists $\epsilon > 0$ such that the proposed preconditioning matrices $\mathbf{A}_{rms,t}$ (Eq. 3) are uniformly equivalent, i.e. Def. 1 holds. The same holds for the diagonal variant.*

---

Consequently, the ellipsoids $\mathbf{A}_{rms,t}$ can directly be applied to any convergent TR framework without losing the guarantee of convergence (Conn et al. (2000), Theorem 6.6.8).[6] In Theorem 1 we extend this result by showing the (to the best of our knowledge) first convergence *rate* for ellipsoidal TR methods. Interestingly, similar results cannot be established for $\mathbf{A}_{ada,t}$, which reflects the widely known vanishing stepsize problem that arises since squared gradients are continuously added to the preconditioning matrix. At least partially, this effect inspired the development of RMSprop (Tieleman & Hinton, 2012) and Adadelta (Zeiler, 2012).

---

[5] In the following we only treat TR methods, but we emphasize that the use of matrix induced norms can directly be transferred to the cubic regularization framework.

[6] Note that the assumption of bounded batch gradients, i.e. smooth objectives, is common in the analysis of stochastic algorithms (Allen-Zhu, 2017; Defazio et al., 2014; Schmidt et al., 2017; Duchi et al., 2011).

## 4.2 A STOCHASTIC ELLIPSOIDAL TR FRAMEWORK FOR NEURAL NETWORK TRAINING

Since neural network training often constitutes a large-scale learning problem in which the number of datapoints $n$ is high, we here opt for a stochastic TR framework in order to circumvent memory issues and reduce the computational complexity. To obtain convergence without computing full derivative information, we first need to assume sufficiently accurate gradient and Hessian estimates.

**Assumption 1** (Sufficiently accurate derivatives)*. The approximations of the gradient and Hessian at step $t$ satisfy*

$$\|\mathbf{g}_t - \nabla \mathcal{L}(\mathbf{w}_t)\| \leq \delta_g \text{ and } \|\mathbf{B}_t - \nabla^2 \mathcal{L}(\mathbf{w}_t)\| \leq \delta_H,$$

*where $\delta_g \leq \frac{(1-\eta)\epsilon_g}{4}$ and $\delta_H \leq \min\left\{\frac{(1-\eta)v\epsilon_H}{2}, 1\right\}$, for some $0 < v < 1$.*

For finite-sum objectives such as Eq. (1), the above condition can be met by random sub-sampling due to classical concentration results for sums of random variables (Xu et al., 2017b; Kohler & Lucchi, 2017; Tripuraneni et al., 2017). Following these references, we assume access to the full function value in each iteration for our theoretical analysis but we note that convergence can be retained even for fully stochastic trust region methods (Gratton et al., 2017; Chen et al., 2018; Blanchet et al., 2016) and indeed our experiments in Section 5 use sub-sampled function values due to memory constraints. Secondly, we adapt the framework of Yao et al. (2018); Xu et al. (2017b), which allows for cheap inexact subproblem minimization, to the case of iteration-dependent constraint norms (Alg. 1).

---

**Algorithm 1** Stochastic Ellipsoidal Trust Region Method

1: **Input:** $\mathbf{w}_0 \in \mathbb{R}^d, \gamma > 1, 1 > \eta > 0, \Delta_0 > 0$
2: **for** $t = 0, 1, \ldots,$ until convergence **do**
3:     Compute approximations $\mathbf{g}_t$ and $\mathbf{B}_t$.
4:     **If** $\|\mathbf{g}_t\| \leq \epsilon_g$, set $\mathbf{g}_t := 0$.
5:     Set $\mathbf{A}_t := \mathbf{A}_{rms,t}$ or $\mathbf{A}_t := \text{diag}\left(\mathbf{A}_{rms,t}\right)$ (see Eq. (3)).
6:     Obtain $\mathbf{s}_t$ by solving $m_t(\mathbf{s}_t)$ approximately.
7:     Compute ratio of function over model decrease:  $\rho_t = \dfrac{\mathcal{L}(\mathbf{w}_t) - \mathcal{L}(\mathbf{w}_t + \mathbf{s}_t)}{m_t(\mathbf{0}) - m_t(\mathbf{s}_t)}$
8:     Set

$$\Delta_{t+1} = \begin{cases} \gamma\Delta_t & \text{if } \rho_{\mathcal{S},t} \geq \eta \\ \Delta_t/\gamma & \text{if } \rho_{\mathcal{S},t} < \eta \end{cases}, \mid : \text{ and } \mathbf{w}_{t+1} = \begin{cases} \mathbf{w}_t + \mathbf{s}_t & \text{if } \rho_t \geq \eta & \text{(successful)} \\ \mathbf{w}_t & \text{otherwise} & \text{(unsuccessful)}. \end{cases}$$

9: **end for**

---

**Assumption 2** (Approximate model minimization)*. Each update step $\mathbf{s}_t$ yields at least as much model decrease as the Cauchy- and Eigenpoint simultaneously, i.e. $m_t(\mathbf{s}_t) \leq m_t(\mathbf{s}_t^C)$ and $m_t(\mathbf{s}_t) \leq m_t(\mathbf{s}_t^E)$, where $\mathbf{s}_t^C$ and $\mathbf{s}_t^E$ are defined in Eq.(28).*

Given that the adaptive norms induced by $\mathbf{A}_{rms,t}$ satisfy uniform equivalence as shown in Lemma 2, the following Theorem establishes an $\mathcal{O}\left(\max\left\{\epsilon_g^{-2}\epsilon_H^{-1}, \epsilon_H^{-3}\right\}\right)$ worst-case iteration complexity which effectively matches the one of Yao et al. (2018).

---

**Theorem 1** (Convergence rate of Algorithm 1)*. Assume that $\mathcal{L}(\mathbf{w})$ is second-order smooth with Lipschitz constants $L_g$ and $L_H$. Furthermore, let Assumption 1 and 2 hold. Then Algorithm 1 finds an $\mathcal{O}(\epsilon_g, \epsilon_H)$ first- and second-order stationary point in at most $\mathcal{O}\left(\max\left\{\epsilon_g^{-2}\epsilon_H^{-1}, \epsilon_H^{-3}\right\}\right)$ iterations.*

---

The proof of this statement is a straight-forward adaption of the proof for spherical constraints, taking into account that the guaranteed model decrease changes when the computed step $s_t$ lies *outside* the Trust Region. Due to the uniform equivalence established in Lemma 2, the altered diameter of the trust region along that direction and hence the change factor is always strictly positive and finite.

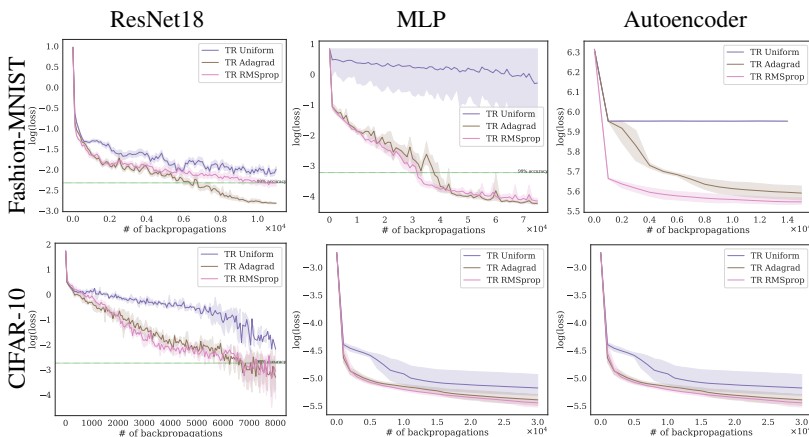

Figure 3: Mean and 95% confidence interval of 10 runs. Green dotted line indicates 99% training accuracy.

# 5 EXPERIMENTS

To validate our claim that ellipsoidal TR methods yield improved performance over spherical ones, we run a set of experiments on two image datasets and three types of network architectures. All methods run on (almost) the same hyperparameters across all experiments (see Table 1 in Appendix B)As depicted in Fig. 3, the ellipsoidal TR methods consistently outperform their spherical counterpart in the sense that they reach full training accuracy substantially faster on all problems. Moreover, their limit points are in all cases lower than those of the uniform method. Interestingly, this makes an actual difference in the image reconstruction quality of autoencoders (see Figure 12), where the spherically constrained TR method struggles to escape a saddle. We thus draw the clear conclusion that the ellipsoidal constraints we propose are to be preferred over spherical ones when training neural nets with second-order methods. More experimental and architectural details are provided in App. C.

To put the previous results into context, we also benchmark several state-of-the-art gradient methods. For a fair comparison, we report results in terms of number of backpropagations, epochs and time. All figures can be found in App. C. Our findings are mixed: For small nets such as the MLPs the TR method with RMSProp ellipsoids is superior in all metrics, even when benchmarked in terms of time. However, while Fig. 9 indicates that ellipsoidal TR methods are slightly superior in terms of backpropagations even for ResNets and Autoencoders, a close look at Fig. 10 and 11 reveals that they at best manage to keep pace with first-order methods in terms of epochs and are inferior in time.

# 6 CONCLUSION

We investigated the use of ellipsoidal trust region constraints for neural networks. We have shown that the RMSProp matrix satisfies the necessary conditions for convergence and our experimental results demonstrate that ellipsoidal TR methods outperform their spherical counterparts significantly across a large set of experiments. We thus consider the development of further ellipsoids that can potentially adapt even better to the loss landscape such as e.g. (block-) diagonal hessian approximations (e.g. Bekas et al. (2007)) or approximations of higher order derivatives as an interesting direction of future research.

Interestingly, the gradient method benchmark indicates that the value of Hessian information for neural network training is limited for mainly three reasons: 1) second-order methods rarely yield better limit points, which suggests that saddles and spurious local minima are not a major obstacle in modern day architectures; 2) The per-iteration time complexity is noticeably lower for first-order methods (Figure 11). The latter observations suggests that advances in distributed second-order algorithms (e.g., Osawa et al. (2018); Dünner et al. (2018)) constitute a promising direction of research towards the goal of a more widespread use of Newton-type methods in deep learning.

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
