# OpenReview forum: "Adaptive norms for deep learning with regularized Newton methods"
_ICLR.cc/2021/Conference — Reject_

### Official Review · AnonReviewer2 · 2020-10-24
**first work studying systematically the practicality of stochatsic ellipsoidal TR method in deep learning**

**Rating:** 6
**Confidence:** 2

**Review:**

The paper proposes novel stochastic ellipsoidal trust-region methods  inspired by adaptive gradient methods and studies the application of them with adaptive diagonal preconditioners. Theoretical convergence analysis is provided for TR with RMSProp ellipsoid, and numerical results demonstrates the superiority of ellipsoided TR over uniform TR. Interestingly, the paper shows for the first time that, adaptive gradient methods can be view as first-order TR with ellipsoidal constraints. The negative comparative results with state-of-the-art adaptive methods are appreciated, showing that the TR-type methods may not be great choices for deep network training, since the Hessians are often diagonal-dominant in deep-learning practice, and hence the benefit of second-order methods are limited.

As said, the paper is mostly well-written and present an insightful investigation of stochastic ellipsoidal TR which could be potentially an alternative to state-of-the-art adaptive gradient methods for modern deep network training, and the paper gives a negative answer. On the other hand, the reviewer feels that the potential impact for the deep learning practitioners may be a bit limited and is not very sure about whether the contribution of this work is that significant.

The experimental details seem unclear. In the experiments, how were the approximate hessian B_t calculated? Are they computed on the same sampled minibatch of gradient or on a new/bigger minibatch? Meanwhile, only training accuracies are reported, which seems inadequate -- should also include the test accuracy plots.

---

> ### Author Response · Authors · 2020-11-12
> **Sorry for the confusion..**
>
> g and B are indeed the sub-sampled gradient and Hessian and they are computed on the same mini-batch! We will clarify this in Appendix C!
>
> Regarding generalization error, we repeat from above: We do not show any test results because this is not what the optimizers are tasked with. Generalization is a very nebulous task and most algorithms that are better at generalization are so because they are worse in training. In our mind, such a comparison is irrelevant unless one presents an algorithm specifically targeted at generalization.
>
> Impact: You are right, we do indeed consider the *currently possible* contribution of second order methods as marginal for the reasons given in the paper but as we also state in the outlook, we do foresee much more frequent use in the future when hardware has advanced.
>
> Finally, we want to thank you for you review and for sharing your concerns!

---

### Official Review · AnonReviewer4 · 2020-10-24
**The paper investigates the use of scaled norms in the constraint of the trust region method.**

**Rating:** 4
**Confidence:** 4

**Review:**

-In section 3.2, the authors use the ratio of the diagonal to the overall mass of a matrix to "measure" the quality of diagonal vs full preconditioning. Why this a good measure? motivation and comments about this measure are needed. I think the most important thing to check here is the "difference" between A^{-1}g and diag(A)^{-1}g. I think an interesting question that one may investigate here is to link this ratio to the  "difference" between A^{-1}g and diag(A)^{-1}g....Or to link this ratio to the complexity...

-In proposition 1, define sigma_1 & 2. I understand that they are defined in the appendix. Just move eq (42) to prop 1. The same about the relation between H & X in proposition 2.

- Rewrite proposition 2, it is a bit misleading, the limit when n goes to infinite should be independent of n ! you may write proportional instead of limit...

-Figure D.2 shows that sqrt(n)/(sqrt(n)+...) decreases to zero with n going to infinite. However, this quantity is non-decreasing as a function of n and it converges to 1 when n diverges!!

-Page 6 after lemma 2, the authors stated that Thm 1 shows the first convergence rate for ellipsoidal TR methods. I disagree about this. The authors may check for instance
the work of Conn et al. (this work is cited in the paper) section 6.7  and Bergou et al.: https://link.springer.com/article/10.1007/s10589-017-9929-2

-  Assumption1: the authors stated that "For finite-sum objectives such as Eq. (1), the above condition can be met by random sub-sampling due to classical concentration results for sums of random variables" this is incorrect! take for instance the case where in the finite sum only one term is not zero and all the other are equal to zero. If the non-zero term is not in the sub-sampling for all t the bounds you mentioned may not be satisfied. You can have these bounds but only in a probabilistic manner as in the Blanchet et al. work (this work is cited in the paper)...

---

> ### Author Response · Authors · 2020-11-12
> **Thank you, please see details below**
>
> First off, thanks again for your detailed and valuable comments.
>
> 3.2 measure of diagonal dominance. We agree that what matters is the closeness of  A^{-1}g and diag(A)^{-1}g. However, note that g is not known a-priori and changes during optimisation as well as from task to task. If A is, however, diagonal, this closeness is guaranteed for all possible vectors g in R^d. That's why!
>
> Proposition 1. Right, we will do so!
>
> Proposition 2 and Fig 6. Very good point Thanks, we will correct that!
>
> Page 6 after Lemma 2: We think that the Conn book only proves convergence and no rate (correct me if I'm wrong) but Bergou et al. is a great pointer. Thank you very much. We'll correct that too!
>
> Assumption 1: This is a very subtle point. Please note that we mention concentration inequalities so it should be clear from the context that we are talking about high probability statements here.

---

### Official Review · AnonReviewer3 · 2020-10-25
**A bridge between adaptive optimization and second order trust region methods**

**Rating:** 5
**Confidence:** 3

**Review:**

This paper analyzes adaptive methods like Adam and AMSProp, and shows that they can be re-interpreted as first order trust region methods with an ellipsoidal trust region (Lemma 1).  The authors then propose a second order trust region method with similar ellipsoidal trust regions induced by the RMSProp matrices (Eq 7). Under some assumptions, they show that this algorithm will converge in a finite number of steps (depending upon the accuracy desired). They also show some experiments to demonstrate their algorithm.

The approach proposed in the paper is interesting, but the significance of the paper is not clear. The application of ellipsoidal trust region to Newton algorithms can certainly help with the development of new optimizers, but the presented algorithm was not very clear to me: I assume approximate g_t is the minibatch gradient, but how is the approximate B_t computed. How is m_t(s_t) computed approximately? Could you say a little more about Assumptions 1 and 2 - when do they hold? Finally, the experimental results were not very clear - it seems like ellipsoidal TR methods are often outperformed by first order methods. Also no generalization results on test sets were shown.

---

> ### Author Response · Authors · 2020-11-12
> **Please find the details in Appendix B and C**
>
> Dear Reviewer
>
> thank you for sharing your thoughts. As we say "g and B are either ∇L(w) and ∇^2L(w) or suitable approximations"
>
> The convergence guarantees hold as long as Assumption 1 is satisfied (which is standard, see the references as well as the references therein).  As stated right after the assumption this can be done via sub-sampling due to concentration inequalities. The result constitutes a non-asymptotic bound on the deviation of the gradient/Hessian norms that holds with high probability.
>
> For our experiments, however, w e use a fixed sample size for the sake of simplicity as stated in Appendix C.
>
> Assumption 2 is always satisfied. Please see Section B2 "Subproblem solver" where we explain that we solve the model using Steihaug-Toint CG, which usually converges in just a few steps but is guaranteed to satisfy A2 after at most d steps. Please see Carmon & Duchi: Analysis of Krylov Subspace Solutions of Regularized Nonconvex Quadratic Problems for more details on the convergence rate of krylov subspace methods on objectives like our model.
>
> Generalization error: We do not show any test results because *this is not what the optimizers are tasked with*. Generalization is a very nebulous task and most algorithms that are better at generalization are so because they are worse in training. In our mind, such a comparison is irrelevant unless one presents an algorithm specifically targeted at generalization.

---

### Official Review · AnonReviewer1 · 2020-10-28
**A different lens to look at second order methods**

**Rating:** 4
**Confidence:** 4

**Review:**

Authors propose a new perspective on adaptive gradient methods. Main contribution is a trust region based algorithm they call "Stochastic Ellipsoidal Trust Region Method" thats flexible to include both full, and diagonal matrix as the preconditioning matrix.  Authors also mention that the preconditioners are generally diagonally dominant in practice, and may only require diagonal matrix (leaves full matrix for future work).

Reason to score:

Weak emperical results, small models, on small datasets without normalizing for batch sizes between experiments.

I have listed my concerns below and hopefully authors can address my concern during the rebuttal period.

I think the authors could substantially improve the emperical results in the paper by including commonly used adaptive methods as baseline (such as Adam), and providing results on stronger baselines, and break down on computational effeciency of the proposed approach in more details.

Questions/comments:

a) There is Appendix C that states that batch size used first order method is 32, vs for this method authors use 128/512 and then compare backprops. This extremely problematic when using # backprop as a way to measure efficiency, as this gives ~4-16x improvement from just larger batch sizes.  I would suggest redoing experiments with exact same batch sizes?

b) Authors indicate using diagonal preconditioner; Could authors consider previous work on kronecker factored preconditioners such as KFAC, or Shampoo that is computationally cheaper in their experiments?

c) Could authors also include walltime comparisons, to split time spent in forward, backward,  hessian-vector product, cg iterations (including details on these as layer sizes increases, for say upto 4k which are common in deep networks trained today?)

d) Could you run a comparison against baselines and settings in: http://www.cs.toronto.edu/~jmartens/docs/Deep_HessianFree.pdf

---

> ### Author Response · Authors · 2020-11-12
> **Epoch results in Appendix Fig 10**
>
> Dear reviewer
>
> thank you very much for sharing your concerns.
>
> Question a) The reasons for picking a higher batch size for second orders methods are detailed in footnote 12 (p.26). This is very standard. Furthermore, we do report results over epochs in Appendix Fig. 10. Please take a look.
>
> Question d) We are comparing the MNIST autoencoder from that paper which is a common benchmark  (Hinton 2006, Xu 2017a, Martens 2010, Martens 2015). Honestly, the other settings seem outdated to us and so does pre-training.
>
> Question b) and c) Unfortunately (?), we are researchers at a public institution and hence we do neither have access to infinite compute power nor do we have a team of research engineers at hand ready to code up whatever experiment setting one can think of.  Note that, neither K-FAC nor Shampoo are available in torch.optim or tensorflow.keras.optimizers.
>
> Benchmarking K-FAC and/or using the K-FAC preconditioner within our framework is a very valuable suggestion though. Thank you! Out of curiosity, why would you consider Shampoo worth benchmarking?

---

> > ### Comment · AnonReviewer1 · 2020-11-18
> > **Baselines should be improved.**
> >
> > Thanks to the authors for their comments.
> >
> > My main concern is still lack of good baseline to compare against existing work, specifically because of changes in critical hyper-parameter:  batch size when comparing various methods.  Justification in the footnote 12, pg. 26. does not explain the discrepancy of why first order methods did not use 128/512 in the experiments. Rerunning these experiments with exact same batch size is recommended.
> >
> > Autoencoder problem on MNIST is a common baseline for comparison and does not require more than a single GPU to compare against KFAC or Shampoo (both are kronecker factored preconditioners).   For example, recent work has even reused them for comparisons: https://arxiv.org/abs/2006.08877 Code also seems to be available to run also. It would help us understand how well the method works against existing work in known settings.  There also seems to be several KFAC baselines availabe in torch, and tensorflow on github.

---

### Decision · Program_Chairs · 2021-01-07
**Final Decision**

**Decision:**

Reject

**Comment:**

The paper considers adaptive stochastic optimization methods and shows that they can be re-interpreted as first order trust region methods with an ellipsoidal trust region, they consider a related second order method, and they show convergence properties and empirical results.

The results are of interest, but the significance of some of the results is not clear.  Part of this has to do with substance, and part of this has to do with presentation that can be improved.  Empirical results are weak, including appropriate baselines and details of the empirical results.